# CFTR Modulator Therapy with Lumacaftor/Ivacaftor Alters Plasma Concentrations of Lipid-Soluble Vitamins A and E in Patients with Cystic Fibrosis

**DOI:** 10.3390/antiox10030483

**Published:** 2021-03-19

**Authors:** Olaf Sommerburg, Susanne Hämmerling, S. Philipp Schneider, Jürgen Okun, Claus-Dieter Langhans, Patricia Leutz-Schmidt, Mark O. Wielpütz, Werner Siems, Simon Y. Gräber, Marcus A. Mall, Mirjam Stahl

**Affiliations:** 1Division of Pediatric Pulmonology & Allergy and Cystic Fibrosis Center, Department of Pediatrics III, University of Heidelberg, Im Neuenheimer Feld 430, 69120 Heidelberg, Germany; olaf.sommerburg@med.uni-heidelberg.de (O.S.); Susanne.Haemmerling@med.uni-heidelberg.de (S.H.); 2Member of the German Center for Lung Research (DZL), Translational Lung Research Center Heidelberg (TLRC), 69120 Heidelberg, Germany; Philipp.Schneider@med.uni-heidelberg.de (S.P.S.); Patricia.Leutz@med.uni-heidelberg.de (P.L.-S.); mark.wielpuetz@med.uni-heidelberg.de (M.O.W.); 3Center for Pediatric and Adolescent Medicine, Department of Paediatrics I, Division of Neuropediatrics and Metabolic Medicine and Newborn Screening Center, University Hospital Heidelberg, 69120 Heidelberg, Germany; JuergenGuenther.Okun@med.uni-heidelberg.de (J.O.); Claus-Dieter.Langhans@med.uni-heidelberg.de (C.-D.L.); 4Department of Diagnostic and Interventional Radiology, University Hospital of Heidelberg, 69120 Heidelberg, Germany; 5Clinics for Prevention and Rehabilitation, 38667 Bad Harzburg, Germany; werner.siems@t-online.de; 6Department of Pediatric Pulmonology, Immunology and Critical Care Medicine and Cystic Fibrosis Center, Charite -Universitätsmedizin Berlin, 13353 Berlin, Germany; simon.graeber@charite.de; 7Berlin Institute of Health (BIH), 10178 Berlin, Germany; 8German Center for Lung Research (DZL), Associated Partner Site, 13353 Berlin, Germany

**Keywords:** cystic fibrosis, CFTR modulators, therapy, retinol, vitamin E, hypervitaminosis A

## Abstract

Rationale: Cystic fibrosis (CF), caused by mutations in the cystic fibrosis transmembrane conductance regulator (CFTR) gene, leads to impaired pancreatic function and therefore reduced intestinal absorption of lipids and fat-soluble vitamins especially in patients with CF developing pancreatic insufficiency (PI). Previous studies showed that CFTR modulator therapy with lumacaftor-ivacaftor (LUM/IVA) in Phe508del-homozygous patients with CF results in improvement of pulmonary disease and thriving. However, the effects of LUM/IVA on plasma concentration of the lipid soluble vitamins A and E remain unknown. Objectives: To investigate the course of plasma vitamin A and E in patients with CF under LUM/IVA therapy. Methods: Data from annual follow-up examinations of patients with CF were obtained to assess clinical outcomes including pulmonary function status, body mass index (BMI), and clinical chemistry as well as fat-soluble vitamins in Phe508del-homozygous CF patients before initiation and during LUM/IVA therapy. Results: Patients with CF receiving LUM/IVA improved substantially, including improvement in pulmonary inflammation, associated with a decrease in blood immunoglobulin G (IgG) from 9.4 to 8.2 g/L after two years (*p* < 0.001). During the same time, plasma vitamin A increased significantly from 1.2 to 1.6 µmol/L (*p* < 0.05), however, levels above the upper limit of normal were not detected in any of the patients. In contrast, plasma vitamin E as vitamin E/cholesterol ratio decreased moderately over the same time from 6.2 to 5.5 µmol/L (*p* < 0.01). Conclusions: CFTR modulator therapy with LUM/IVA alters concentrations of vitamins A and vitamin E in plasma. The increase of vitamin A must be monitored critically to avoid hypervitaminosis A in patients with CF.

## 1. Introduction

Cystic fibrosis (CF) is a complex multi-organ disease caused by mutations in the cystic fibrosis transmembrane conductance regulator gene (CFTR gene) [1,2]. The CFTR protein forms a cAMP-regulated ion channel responsible for chloride and bicarbonate secretion in epithelial cells. Since CFTR regulates also the epithelial Na+ channel (ENaC), it plays a central role in the adequate humidification of epithelial surfaces [3]. The CFTR defect in the lungs leads to chronic airway mucus obstruction, infection, and inflammation. The resulting progressive lung damage remains with about 80% the leading cause of morbidity and mortality in CF [4]. Of the about 2100 CFTR mutations known today, the Phe508del mutation is present in about 90% of cases on at least one allele, and nearly half of all CF patients are homozygous for Phe508del [5].

Infants with CF usually develop extrapulmonary symptoms as first signs of disease. A meconium ileus occurs in up to 15% of infants with CF [6]. At a later stage, failure to thrive or a distended abdomen are common due to CF-related exocrine pancreatic insufficiency (PI), which is already present at birth in up to 85% of infants with CF [7,8]. As a result, enzyme- and bicarbonate-containing secretions cannot adequately be secreted into the duodenum, leading to maldigestion and malabsorption, especially of fats and fat-soluble vitamins [9]. CF-related PI is treated by pancreatic enzyme replacement therapy (PERT) with pancreatic lipase-containing preparations (2000−4000 IU of lipase per gram of dietary fat), a high-calorie, high-protein, high-fat diet (130–150% of the energy level of healthy subjects of the same age), and age-adjusted substitution of fat-soluble vitamins A, D, E, and K [10]. Monitoring of serum levels of fat-soluble vitamins A, D, E, K and their individual dose adjustment should be part of regular CF management [11].

In the literature, vitamin A (retinol) deficiency is described in 10–40% of patients with CF, irrespective of whether vitamin A supplementation was performed [12,13,14,15]. Reports of typical symptoms of vitamin A deficiency, such as night blindness, conjunctival and corneal xerosis, blindness and phrynoderma are rare, but impaired immunological host defense and poor thriving is certainly present in a proportion of patients with CF [16,17]. On the other hand, it must be noted that excessive vitamin A intake may lead to hepatotoxicity and lower bone mineral density in patients with CF [18].

Vitamin E is an overarching term for a group of tocopherols and tocotrienols that can act as antioxidants in various ways [19]. It is important for cell membrane integrity and plays a functional role in neuronal tissues. Dietary intake of vitamin E alone is inadequate in CF patients with PI, even under regular PERT treatment [10]. Since vitamin E (α-tocopherol) is transported almost exclusively in circulating lipoproteins in blood, serum levels depend strongly on the amount of circulating lipids, which are often low in patients with CF. A number of older studies showed that patients with CF may have significant vitamin E deficiency, which is more pronounced the longer adequate vitamin E substitution is not provided (e.g., [12,20,21]). However, patients with CF achieve normal serum levels of vitamin E if they receive 5−10 IU/kg/day of vitamin E with food or supplements along with PERT (e.g., [20,22,23,24]).

Recently, small molecule drugs have become available for patients with CF with certain genotypes acting as potentiators and correctors of CFTR function [25,26,27,28]. The combination of the corrector lumacaftor (LUM) with the potentiator ivacaftor (IVA) was the first drug approved to restore mutant Phe508del-CFTR function in homozygous patients with CF [29]. However, compared with the highly effective CFTR modulator therapy with ivacaftor in patients with CF and a Gly551Asp gating mutation or with the recently approved new combination therapy composed of the correctors elexacaftor and tezacaftor and the potentiator ivacaftor, LUM/IVA therapy only showed a moderate effect on lung function with high heterogeneity between patients [29,30,31,32,33]. It was recently demonstrated that LUM/IVA therapy leads to a partial rescue of CFTR function of about 10% to 20%, which is comparable to the lower range of CFTR function in patients with CF and residual function mutations [34].

To the best of our knowledge, no study has yet investigated the course of serum concentrations of fat-soluble vitamins, especially vitamin A and E, over a longer period of time under treatment with a CFTR modulator. In this context, the group of patients with CF homozygous for Phe508del could especially represent a homogeneous population for such a project. Therefore, the aim of our study was to investigate the longitudinal concentration of both supplemented vitamins A and E in children and adolescents with CF homozygous for the CFTR mutation Phe508del before and after at least 12 months of treatment with LUM/IVA.

## 2. Materials and Methods

### 2.1. Study Population

This project was performed as part of a prospective longitudinal observational study in patients with CF followed at the CF Center in Heidelberg, Germany, and was approved by the ethics committee of the University of Heidelberg (S-211/2011). Written informed consent was obtained from all patients, their parents or guardians before initiation. For the present study, we used pseudonymized data from patients with CF homozygous for the CFTR mutation Phe508del, suffering from pancreatic insufficiency (fecal elastase <200 µg/g stool) and treated with the CFTR modulator LUM/IVA (Orkambi^®^, Vertex Inc., Philadelphia, PA, USA) in the last four years. LUM/IVA is approved in Europe for patients with CF two years of age and older, so patients two years of age and older were eligible to participate in the study. Exclusion criteria for this study were severe liver cirrhosis, very poor pulmonary function with a forced expiratory volume in one second in percent predicted (FEV1%pred) <40%, acute exacerbation at the time of examination, and knowledge of poor adherence to medication including PERT and fat-soluble vitamin supplementation. All patients received vitamin supplementation according to current guidelines [10]. Care was taken to ensure that vitamin supplementation was carried out regularly. Variations in the dosage of individual vitamins were only made if the plasma concentrations deviated from the target range (0.70–2.51 µmol/L for plasma vitamin A and 2.5–15.0 µmol/mmol for vitamin E/cholesterol) [16,35] and deficiency symptoms or toxicity could have been expected as a result.

### 2.2. Anthropometry, Lung Function Tests

To describe the clinical condition of CF patients, we compared the results of annual checkups before and after initiation of LUM/IVA and annually thereafter. Thus, theoretically, a maximum follow-up of three years was possible during the study period of four years in a subgroup of patients with CF treated with LUM/IVA. Height and weight were measured during the control examinations using standard techniques. Z-scores for BMI were derived from reference values of healthy children in Germany, the calculations of anthropometric data were performed using the software GrowthXP 2.6 (PC Pal, Bandhagen, Sweden).

Lung function analysis was performed in all CF patients older than four years according to the criteria of the American Thoracic Society and the European Respiratory Society [36]. FEV1%pred was calculated in accordance with the multi-ethnic reference values for spirometry for children and adolescents published by the European Respiratory Society [37]. According to that values for FEV1%pred that are above 80% are considered normal. The lung clearance index (LCI) is a measure of mucus-related ventilation inhomogeneities in the airways. In addition to the spirometrically determined FEV1, the LCI represents another sensitive description of the state of lung disease in CF. The LCI was measured by the so-called multiple breath washout (MBW) test with N2 as tracer gas. The technical MBW procedures were carried out in accordance with the American Thoracic Society Technical Statement and were performed on awake, upright children or adolescents using the commercially available mainstream ultrasonic flowmeter Exhalyzer D and spiroware 3.2.1 as acquisition software (Eco Medics, Duernten, Switzerland) [38,39,40,41]. If necessary, children were distracted by watching a video while breathing tidally through a mouthpiece with a nose clip in place. The 95% limits of normality for the LCI performed with N2 range from 5.3 to 7.3 in children (up to 16 years of age) and from 5.9 to 7.5 in adults [38,39,40,41].

### 2.3. Measurement of Biochemical Routine Parameters, Vitamin A, Vitamin E

Different laboratory parameters were measured together with plasma vitamin A and vitamin E once a year in each CF patient at a time of clinical stability. Plasma and serum samples for these laboratory tests were collected in the morning shortly after admission and C-reactive protein (CrP), immunoglobulin G (IgG), alanine transaminase (ALT), aspartate transaminase (AST), gamma-glutamyltransferase (GGT), alkaline phosphatase (AP), creatinine, albumin, cholesterol, international normalized ratio (INR) of prothrombin time as coagulation parameter as indirect marker for vitamin K, and 25-OH-cholecalciferol (vitamin D) were analyzed immediately using automated biochemistry analyzers in the DIN EN ISO 15189 certified central laboratories of the University Hospital Heidelberg. For the measurement of vitamin A (as retinol) and vitamin E (as α-tocopherol), serum samples were separated from blood cells and then stored at −80 °C. The analysis of vitamin A and vitamin E was performed usually once a week. For extraction 200 µL water was added to 200 µL serum and while vortexing mixed with 400 µL ethanol. After 5 min at room temperature, the mixture was extracted with 1000 µL n-hexane. An aliquot of the solvent (400 µL) was evaporated to dryness in a stream of nitrogen. The residue was dissolved in 200 µL of a mixture of acetonitrile/dioxane/ethanol (3/1/1, *v*/*v*/*v*). For measurement, a Beckman Coulter Gold HPLC pump model 127 with fluorescence detector RF-551 (Shimadzu, Tokyo, Japan) was used. Chromatographic separation was achieved on an Ascentis C18 column (2.7 µm, 4.6 mm × 100 mm, Sigma-Aldrich, St. Louis, MO, USA) with acetonitrile/tetrahydrofuran/methanol/ammoniumacetate (1%) (684/220/68/28, *v*/*v*/*v*/*v*) as mobile phase. The injection volume was 20 µL. The detection of both vitamins was performed fluorometrically. Retinol was detected at λex = 330 nm and λem = 470 nm and α-tocopherol at λex = 298 nm and λem = 330 nm.

### 2.4. Statistics

Data were held in a Microsoft Access^®^ database and are presented as mean ± standard deviation (SD) if they were normally distributed variables. Non-normally distributed variables were presented as median values and minimum and maximum or single values. Comparisons of two groups were performed with a paired *t*-test when normally distributed or Wilcoxon matched-pairs signed rank test when non-normally distributed, respectively. Comparisons of three groups were performed with a one-way ANOVA test with Greenhouse-Geisser correction when normally distributed or Friedman test when non-normally distributed. Statistical significance was accepted at the level of *p* < 0.05. All statistical calculations were performed with Prism 6.07 for Windows (Graph Pad software Inc., San Diego, CA, USA).

## 3. Results

Between December 2016 and December 2020, 55 of 146 patients with CF of our CF center were treated with CFTR modulators (Figure 1). Of these, 45 patients with CF were homozygous for the Phe508del CFTR mutation, at least two years old and treated with LUM/IVA. After reviewing the inclusion and exclusion criteria, four of these patients were excluded, so in the end 41 subjects were enrolled into the study (Figure 1). Anthropometry, spirometry and/or MBW, and monitoring of clinical laboratory parameters, were performed in all patients before and at least one year after initiation of LUM/IVA therapy.

All 41 patients with CF who completed the baseline visit before starting LUM/IVA were followed up for at least one year. The demographic data and patient characteristics of all these patients at baseline are summarized in Table 1. A subgroup of 21 subjects could be followed for another year. The median age of these patients was 7.6 (4.2–14.8) years at baseline, and 11 (52.4%) of them were female. Of these 21 patients with CF, only 10 subjects (6 female, 60.0%) could so far be followed for three years under CFTR modulator therapy, leaving this subgroup too small for reliable statistical conclusions.

### 3.1. Effect of Treatment with LUM/IVA on Anthropometry and Lung Function Tests

Consistent with the results of previous clinical trials, LUM/IVA improved the thriving of patients after one year of treatment. We observed a significant increase in BMI z-score from −0.24 ± 0.88 to −0.04 ± 0.96 (*p* < 0.05). However, there was no general trend in pulmonary function tests after one year of LUM/IVA. While there was no improvement in FEV1%pred, we saw a significant reduction in LCI, which decreased by 1.46 points (*p* < 0.01) after one year of LUM/IVA therapy. All details are given in Table 2.

In the subgroup followed for two years, the BMI z-score increased significantly from −0.15 ± 0.83 to 0.24 ± 0.63 (*p* < 0.001) (Figure 2A). Lung function parameters showed again no effect in FEV1%pred (Figure 2B). Interestingly, although there is a trend toward improvement in LCI over two years, the improved values were no longer statistically significant as seen after one year of LUM/IVA therapy (Figure 2C).

### 3.2. Effect of Treatment with LUM/IVA on Biochemical Parameters, Retinol, and Vitamin E

An overview of the biochemical parameters of the 41 patients with CF in the study at the time before initiation of LUM/IVA therapy is shown in Table 1. All subjects showed predominantly normal values for clinical chemistry. Liver values were on average within the normal range. As expected, elevated but still acceptable values for GGT, AST, ALT, and AP were found in some patients with CF with known CF-related hepatopathy. CrP was normal in most patients at baseline and throughout the study, with only some showing moderate CrP elevations (Table 1, Table 2 and Table 3). IgG was also predominantly within the normal range before LUM/IVA, although some patients showed elevated levels, which is typical in patients with CF who have chronic inflammation (Table 1 and Table 2). Interestingly, after one year of LUM/IVA therapy IgG dropped significantly (Table 2). The same was seen in the subgroup of the 21 CF patients followed over two years (*p* < 0.001) (Table 3). The vitamin D (25-OH-Cholecalciferol) concentration remained unchanged under LUM/IVA therapy. There was a slight upward trend in the median after one year as well as after two years, but the changes were not significant (Table 2 and Table 3). The situation was somewhat different when considering the INR of coagulation as a vitamin K-dependent parameter. Here, after one year, there was a moderate but significant decrease in the INR in the total cohort (*p* < 0.001) (Table 2). However, this trend could no longer be observed in the subgroup of 21 CF patients who were followed up over two years of LUM/IVA therapy (Table 3). Retinol levels increased slightly but not significantly after 1 year of LUM/IVA treatment (Table 2). However, in the 21 CF patients followed for two years of LUM/IVA therapy, serum retinol increased significantly (*p* < 0.05) from 1.23 (0.79–2.27) to 1.60 (0.85–2.49) during this time (Figure 2 and Table 3). Serum cholesterol decreased moderately after initiation of LUM/IVA therapy. As seen in Table 2, after one year of LUM/IVA therapy there was a significant decrease of cholesterol in the entire cohort of 41 CF patients (*p* < 0.05). However, this trend could no longer be observed in the subgroup of 21 CF patients who were followed up over two years of LUM/IVA therapy (Table 3). Serum vitamin E decreased significantly after one (*p* < 0.01) and after two years (*p* < 0.01) of LUM/IVA therapy (Table 2 and Table 3). This was observed for vitamin E alone but also for the more informative vitamin E/cholesterol ratio. The ratio decreased significantly in the 41 CF patients after one year of LUM/IVA therapy (*p* < 0.05), and the same was true for the 21 CF patients treated with LUM/IVA for two years (*p* < 0.05).

## 4. Discussion

Mutation-specific CFTR modulator therapy led to significant clinical improvements in patients with CF. LUM/IVA was the first CFTR modulator combination approved for Phe508del-homozygous patients with CF [42]. Phase 3 trials demonstrated safety and clinical efficacy of LUM/IVA therapy [29,30]. In a study measuring the extent of in vivo correction of CFTR function by using the CFTR biomarkers sweat chloride concentration, nasal potential difference measurement, and intestinal current measurement, LUM/IVA therapy led to an improvement of 10–20% of normal CFTR activity in Phe508del-homozygous patients [34]. The present study demonstrates for the first time that partial restoration of CFTR function in Phe508del-homozygous patients with CF over time also leads to changes in plasma levels of the fat-soluble vitamins A and E, which may be of clinical relevance to patients.

### 4.1. CF Lung Disease under Lumacaftor/Ivacaftor Therapy

In our study, we did not see significant improvements in FEV1%pred as seen in the pivotal studies of LUM/IVA [29,30]. The reason for this is probably that the number of patients considered here is too small to detect such subtle changes. On the other hand, our data in this cohort of patients with well-preserved spirometry are in line with pivotal studies on LUM/IVA in children aged 6 to 11 years that showed no improvement in FEV1%pred [43], as this parameter is not well suited to detect small changes when lung function is still relatively normal in young children with CF. In contrast, we saw a significant improvement in LCI in the first year of LUM/IVA therapy in the cohort of all 41 CF patients studied (Table 2). This is consistent with previous publications [43,44,45]. In our subgroup followed up for two years (*n* = 21), it was shown that the positive effect of LUM/IVA therapy on LCI may even persist after the first year of treatment, although these changes were not statistically significant in this smaller subgroup. This suggests an effective improvement in mucociliary clearance in these patients, resulting in less mucus plugging and thus probably less local inflammation in the airways. This conclusion was supported by the fact that in parallel with LUM/IVA therapy and the described improvements in ventilation homogeneity, there was a marked decrease in total IgG during the first two years of LUM/IVA therapy in the patients with CF. Although only a minority of patients with CF showed elevated IgG at baseline, an increased immunological activity due to chronic inflammation can be assumed as part of the pulmonary CF disease in all patients with CF [46,47]. If a significant IgG decrease is now observed, it can be assumed that the reduced inflammatory activity in these patients with CF is associated with the partial correction of the CFTR defect under LUM/IVA therapy.

### 4.2. Abdominal CF Disease under Lumacaftor/Ivacaftor Therapy

Consistent with the results of previous clinical trials [29,30,34], LUM/IVA improved the thriving of our predominantly young patients with CF. Thus, BMI z-score increased significantly by 0.18 points after one year of LUM/IVA therapy in the cohort of our 41 patients with CF. But even after two years, a significant increase in BMI z-score was still seen. This indicates a substantial improved digestion and absorption of food, especially of fats and of lipid-soluble nutrients.

Although patients with CF have increased cholesterol biosynthesis in the liver [48,49], their plasma cholesterol is nevertheless decreased [50]. The increased biosynthesis does not lead to increased secretion of cholesterol into the blood, but to intrahepatic accumulation apparently due to altered cholesterol trafficking in liver cells, which in turn leads to inflammatory responses [48,49]. In a recent study, it was reported that therapy with LUM/IVA in 20 patients with CF homozygous for Phe508del did not lead to an increase in plasma cholesterol as expected, but on the contrary to a further moderate decrease [51]. Interestingly, our data confirm this observation, at least in the cohort evaluated after one year of LUM/IVA therapy. In the aforementioned work, it was suggested that the reason for the further decrease in plasma cholesterol seen may be normalized hepatic cholesterol biosynthesis, which in combination with the still CFTR-related dysfunctional secretion of cholesterol from the liver into the blood under LUM/IVA therapy leads to the lower plasma concentration [51]. On the other hand, it is possible that more cholesterol is needed for bile acid synthesis when fat absorption is normalized [51]. Interestingly, the detection of cholestanol, a surrogate marker for intestinal absorption [52], indicates increased cholesterol absorption from the intestine [53]. This fact is important as it points in the direction that the absorption of other fat-soluble micronutrients is also improved under LUM/IVA therapy. Despite this assumption, there were no substantial changes in vitamin D concentration. The situation may be somewhat different for vitamin K. Here, we saw a moderate but significant (*p* < 0.001) increase in INR in the first year after LUM/IVA therapy, which could be indicative of slightly impaired coagulation because of a potentially decreased vitamin K concentration. However, because of the many factors that influence coagulation, we cannot draw this conclusion definitively, but on the other hand; we cannot explain the result as well. The facts that the amount of INR increase was not clinically relevant and that this tendency was no longer seen in the subgroup of CF patients followed up two years after the initiation of LUM/IVA therapy highlights the need for follow-up data in a larger patient group and over a longer period.

### 4.3. Vitamin A (Retinol)

Vitamin A is routinely administered to all patients with CF with PI. There are different data about the frequency of vitamin A deficiency in CF [12,13,14]. In recent years, however, an increasing number of studies have been published in which such a deficiency was hardly seen or not seen at all [35,54,55]. In the data collected for our study, we were also unable to detect a deficiency of vitamin A in the plasma of any of the patients with CF at baseline. It can be assumed that the therapy of fat malabsorption is nowadays so effective that vitamin A deficiencies are hardly ever observed in patients with supplementation of lipid-soluble vitamins. After one year of LUM/IVA therapy, there was a trend toward a moderate increase in plasma vitamin A in our cohort of young patients with CF. In the subgroup of patients with CF followed up for two years on LUM/IVA therapy, this increase reached significance (*p* < 0.05, Figure 2). However, none of the patients with CF in our study developed excessive plasma levels of vitamin A. In principle, however, hypervitaminosis A may also occur in patients with CF. In a recent Dutch study monitoring 221 patients with CF longitudinally over seven years, three measurements in three children showed toxic Vitamin A plasma values above 3.5 µmol/L [54]. Therefore, the question arises whether patients on CFTR modulators should be monitored more closely for vitamin A serum levels in order to avoid the development of hypervitaminosis A. In general, liver damage associated with hypervitaminosis A is well known, but it may be clinically overlooked because the biochemical liver dysfunction in this case is often mild and the morphologic features may also be similar to those of CF-related hepatopathy. Supplemented vitamin A consists of preformed retinoids that are solubilized and esterified in the intestine to retinyl esters and then stored in the liver as retinol. Thus, overloading the liver with retinol via excessive vitamin A intake basically carries the risk of hepatotoxicity. The spectrum of liver disease ranges from non-cirrhotic portal hypertension to fibrosis or even cirrhosis. Data from the literature indicate that even prolonged and continuous consumption of vitamin A doses in the low “therapeutic” range can already lead to life-threatening liver damage [56]. Since there is insufficient evidence for the vitamin A doses recommended for CF patients today, the lowest vitamin A dose that can achieve sufficient plasma concentrations should always be selected [10,55]. This is especially true after starting CFTR modulator therapy, which on the one hand improves the malabsorption of fats and thus the absorption of vitamin A, but probably much more reduces the number of acute lung exacerbations and thus improves the chronic inflammatory state of the lungs in the long term, which in turn would reduce the need for retinol. Interestingly, a recent paper reported an inverse correlation between serum vitamin A concentration and IgG as a marker for chronic inflammatory conditions [55]. Due to the limited number of patients in our cohort and because such an analysis would have been beyond the scope of our study, we did not perform statistics on this, but we were also able to observe this trend (Figure 2D,E). However, the extent to which acute infections in our CF patients may lead to temporarily decreased plasma vitamin A deficiency, if any, cannot be answered in this study because CF patients with acute infections were excluded. However, there are estimates describing that infections may lead to a decrease in plasma retinol even in healthy individuals.

### 4.4. Vitamin E

Following guidelines-based oral supplementation of vitamin E, deficiency seems to be rare in patients with CF today. Similar to retinol, the risk of vitamin E deficiency increases with inflammation of the respiratory or digestive system [14,57]. On the other hand, excessive supplementation of vitamin E may lead to high serum levels of α-tocopherol and possible toxicity [58,59]. Overall, however, the evidence documenting the potential predictors of vitamin E levels in CF patients is sparse [10,58]. In our study, we observed that plasma vitamin E decreased moderately but significantly under CFTR modulator therapy. This effect was evident not only after one year, but also when the corresponding subgroup of patients with CF was evaluated after two years of LUM/IVA treatment. Moreover, this effect was observed both when plasma vitamin E concentrations were considered individually and when the ratio of vitamin E and cholesterol concentrations was used. However, the decrease in plasma vitamin E concentrations did not lead to vitamin E deficiency either clinically or in terms of laboratory normal values. This result contrasts with the aforementioned cholesterol study, which also examined vitamin E in CF patients before and nine months after initiation of LUM/IVA therapy. The authors of this study described a moderate vitamin E increase [51]. However, the decreasing vitamin E levels found in our study may have various reasons. One reason could be that vitamin E competes with other fat-soluble substances when absorbed from the intestine. In previous studies, it was shown that the absorption of vitamin E depends largely on its distribution in the intestinal lumen [60] and that oral intake of sterols, stanols, and LDL cholesterol resulted in a moderate reduction in plasma vitamin E concentrations, but without clinical effects [61]. In addition, a clinical study showed that CFTR modulators normalize intestinal pH in the long term, which is also expected to alter fat digestion in the intestine [62]. A third reason for a slightly decreased plasma vitamin E concentration could be an altered vitamin E distribution in the body tissues of CF patients. It is to be expected that the observed increase in BMI is also accompanied by an increase in adipose tissue in these patients, where vitamin E is accumulated to a considerable extent. Thus, it is also possible that supplemented vitamin E is redistributed via LDL from the plasma into the adipose tissue that is now present in the body [63]. A fourth reason could be the drug lumacaftor itself, as this is a known inducer of cytochrome P450 3A4, which in turn would also affect the degradation of vitamin E [64]. Based on the different results, no general conclusions can be drawn as to how the plasma concentration of vitamin E is affected by CFTR modulator therapy. Therefore, the current management of vitamin supplementation and monitoring of plasma concentrations remains, at least for CFTR modulator therapy with LUM/IVA.

### 4.5. Limitations

Our study has some limitations. For example, we know the vitamin dosage prescribed to each of our patients, but we have not collected reliable data from which the real intake of vitamins A and E could be evaluated. On the other hand, keeping dietary records may be so burdensome for patients whose time is already limited by therapy that over- and/or under-reporting may occur, which in turn may have affected the validity of the data. It was also not taken into account whether the patients took vitamin A enriched food. However, experience has shown that vitamin A enriched foods are unlikely to lead to higher vitamin A plasma levels in our study, as the vitamin A content in these foods is limited to a very low level in Germany.

In order to correctly assess the relevance of retinol in plasma, the retinol binding protein should also be determined, but this is not included in the routine monitoring of CF patients. In principle, plasma retinol as a parameter for vitamin A supply should also be questioned. It is well known that plasma retinol gives good information about vitamin deficiencies. For the assessment of hypervitaminosis A, however, other parameters, for example also the plasma concentrations of retinyl esters, would have to be determined in future studies.

## 5. Conclusions

Patients with CF homozygous for the Phe508del mutation who received LUM/IVA therapy improved significantly in pulmonary function and thriving. In addition, blood IgG decreased significantly as a sign of reduced lung inflammation. In parallel, plasma vitamin A increased, but no patient with CF showed toxic levels. In contrast, plasma vitamin E decreased moderately during the same period. It should be noted that the increase in vitamin A must be critically monitored to avoid hypervitaminosis A in patients with CF receiving CFTR modulator therapy. Since newer CFTR modulators such as elexacaftor/tezacaftor/ivacaftor exceed the efficacy of LUM/IVA several times over, even more caution is needed with these newer CFTR modulators regarding a balanced vitamin supplementation.

## Figures and Tables

**Figure 1 antioxidants-10-00483-f001:**
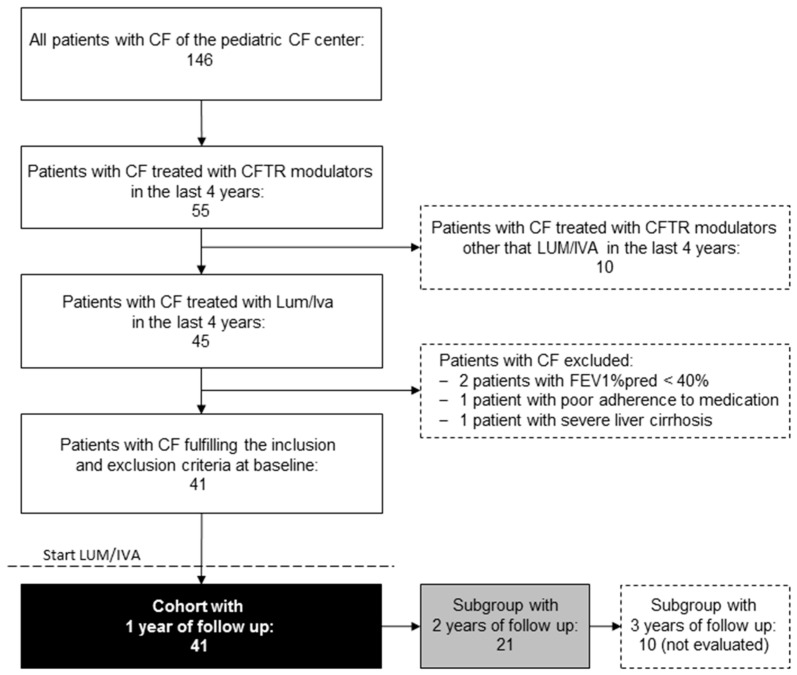
Flow chart showing patient recruitment.

**Figure 2 antioxidants-10-00483-f002:**
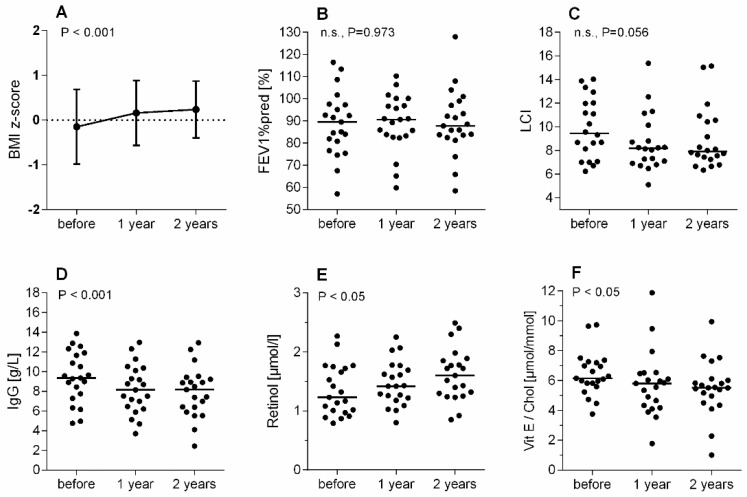
Course of selected clinical parameters and fat-soluble vitamins over two years of therapy with lumacaftor/ivacaftor (LUM/IVA) in a cohort of 21 patients with CF. (**A**) BMI z-scores given as mean ± SD. (**B**) Forced expiratory volume in one second in percent predicted (FEV1%pred), (**C**) Lung clearance index (LCI), (**D**) immunoglobulin G (IgG), (**E**) retinol, and (**F**) ratio of vitamin E and cholesterol. Values in (**B**–**F**) are presented as single values and median. Results with *p* < 0.05 are statistically significant. Abbrevations: n.s. = not significant.

**Table 1 antioxidants-10-00483-t001:** Characteristics at baseline in 41 patients with cystic fibrosis (CF).

Clinical Parameter	Median (Range)
Age [years]	6.6 (2.1–21.8)
Sex, female, *n* (%)	20 (48.8%)
CFTR-Genotype: F508del/F508del, *n* (%)	41 (100%)
Pancreatic insufficiency (PI), *n* (%)	41 (100%)
BMI [kg/m^2^]	15.8 (13.7–23.47)
Albumin [g/L]	45.1 (29.4–50.2)
Creatinine [mg/dL]	0.34 (0.19–0.74)
CrP [mg/L]	2.0 (2.0–17.2)
IgG [g/L] (*n* = 39)	9.36 (4.75–21.79)
GGT [U/L]	11.0 (7.0–36.0)
AST [U/L]	31.0 (12.0–111.0)
ALT [U/L]	24 (1.7–190)
AP [U/L]	287 (109–475)

Definition of abbreviations: BMI = body mass index; FEV1%pred = forced expiratory volume in one second in percent predicted; CrP = C-reactive protein; GGT = gamma-glutamyltransferase; AST = aspartate transaminase; ALT = alanine transaminase; AP = alkaline phosphatase. If dataset was incomplete for a certain parameter, the number of patients for the respective parameter is given in parenthesis.

**Table 2 antioxidants-10-00483-t002:** Course of clinical parameters and fat-soluble vitamins after one year of therapy with lumacaftor/ivacaftor (LUM/IVA) in a cohort of 41 CF patients.

	Before Start LUM/IVA	1 Year after Start LUM/IVA	*p* Value
BMI z-score (mean ± SD)	−0.24 ± 0.89	−0.04 ± 0.96	0.0432 *
FEV1%pred (*n* = 34)	84.9 (40.6–122.8)	87.6 (33.8–110.3)	0.8428
LCI (*n* = 39)	9.00 (5.63–21.12)	7.54 (5.11–21.58)	0.0018 **
CrP [mg/L]	2.0 (2.0–17.2)	2.0 (2.0–47.0)	0.7422
IgG [g/L] (*n* = 39)	9.36 (4.75–21.79)	8.13 (3.19–19.17)	0.0001 ***
INR (*n* = 39)	1.01 (0.94–1.25)	1.06 (0.93–1.48)	0.0001 ***
25-OH-Cholecalciferol [nmol/L] (*n* = 40)	63.9 (22.0–154.0)	76.3 (23.0–134.5)	0.1954
Retinol [µmol/L]	1.23 (0.51–2.50)	1.33 (0.34–2.56)	0.3874
Vitamin E [µmol/L]	22.0 (12.7–39.7)	19.5 (4.8–45.6)	0.0020 *
Cholesterol [mmol/L]	3.36 (2.15–4.88)	3.27 (1.51–4.45)	0.0209 *
Vitamin E/Cholesterol [µmol/mmol]	6.45 (3.43–11.23)	5.96 (1.78–11.88)	0.0152 *

Definition of abbreviations: BMI = body mass index; FEV1%pred = forced expiratory volume in one second in percent predicted; LCI = lung clearance index; IgG = immunoglobulin G; INR = international normalized ratio of prothrombin time. If dataset was incomplete for a certain parameter, the number of patients for the respective parameter is given in parenthesis. Values are given as median (range), if not indicated otherwise. * *p* < 0.05, ** *p* < 0.01, *** *p* < 0.001.

**Table 3 antioxidants-10-00483-t003:** Course of clinical parameters related to fat-soluble vitamins after two years of therapy with lumacaftor/ivacaftor (LUM/IVA) in a cohort of 21 patients with CF.

	Before Start LUM/IVA	1 Year after Start LUM/IVA	2 Year after Start LUM/IVA	*p* Value
Patient number, N cohort (N female, %)	21			
	(11, 52.4%)			
Age at start of LUM/IVA (years)	7.6			
	(4.2–14.8)			
BMI z-score (mean ± SD)	−0.14 ± 0.83	0.16 ± 0.73	0.24 ± 0.63	0.0003 ***
FEV1%pred	89.6	90.7	87.4	0.9731
	(57.1–116.5)	(59.9–110.3)	(58.2–128.0)	
LCI (*n* = 20)	9.44	8.19	7.92	0.0556
	(6.24–14.03)	(5.11–15.39)	(6.34–15.15)	
CrP [mg/L]	2.0	2.0	2.0	0.3679
	(2.0–2.0)	(2.0–9.0)	(2.0–13.2)	
IgG [g/L]	9.36	8.13	9.36	0.0006 ***
	(4.75–13.87)	(3.70–13.87)	(4.75–12.92)	
INR (*n* = 19)	1.02	1.01	1.01	0.4791
	(0.96–1.18)	(0.97–1.14)	(0.98–1.13)	
25-OH-Cholecalciferol [nmol/L] (*n* = 20)	61.0	67.9	73.1	0.3160
	(22.0–118.3)	(23.0–125.0)	(19.0–134.8)	
Retinol [µmol/L]	1.23	1.42	1.60	0.0140 *
	(0.79–2.27)	(0.80–2.25)	(0.85–2.49)	
Vitamin E [µmol/L]	20.9	18.2	16.7	0.0085 **
	(14.5–31.9)	(5.58–33.4)	(3.2–30.5)	
Cholesterol [mmol/L]	3.43	3.20	3.21	0.0945
	(2.51–4.13)	(2.04–4.22)	(1.91–4.18)	
Vitamin E/Cholesterol [µmol/mmol]	6.15	5.81	5.52	0.0268 *
	(3.75–9.74)	(1.78–11.88)	(1.00–9.95)	

Definition of abbreviations: BMI = body mass index; FEV1%pred = forced expiratory volume in one second in percent predicted; LCI = lung clearance index; IgG = immunoglobulin G; INR = international normalized ratio of prothrombin time. If dataset was incomplete for a certain parameter, the number of patients for the respective parameter is given in parenthesis. Values are given as median (range), if not indicated otherwise. * *p* < 0.05, ** *p* < 0.01, *** *p* < 0.001.

## Data Availability

Data is contained within the article.

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
