# Peer review of "CFTR Modulator Therapy with Lumacaftor/Ivacaftor Alters Plasma Concentrations of Lipid-Soluble Vitamins A and E in Patients with Cystic Fibrosis"

_antioxidants, 2021, doi:10.3390/antiox10030483_

Round 1

Reviewer 1 Report

This report by Sommerburg et al describes a valuable study that may be important for the future of vitamin supplementation for individuals with CF in the context of CFTR modulator treatment. The manuscript is straightforward, and I only have a few concerns.

Concerns:

1) It is stated in the methods that “Variations in the dosage of individual vitamins were only made if the plasma concentrations deviated from the target range…”. Were changes made in the dosage of vitamin supplementation over the course of LUMA/IVA treatment? If so, how common were these changes? If these were common, it would be important to mention this in the manuscript as another potential confounder of the results.

2) The statistical analyses performed for Figure 2 and those in the text describing Figure 2 and Table 3 are not clear to this reviewer. In lines 236 – 238, the loss of a significant reduction in LCI over two years is described, but in Figure 2C, I believe that the p value shown is for an ANOVA/Friedman test, which would just describe differences amongst the groups, but these tests would not make comparisons between two groups (unless a post-test was also performed). Were paired t-test/Wilcoxon matched-pairs signed rank tests performed between the “before” vs. “1 year” (and was this significant for the 21 patients?) and also between “before” vs. “2 years” (no longer significant)? I assume this is what was performed for the p values described in lines 271 – 275. While the p values provided by ANOVA/Friedman tests are meaningful to describe general differences amongst groups, it could be that no significant difference was observed between the “before” and “Year 1”, but a difference was observed between “Year 1” and “Year 2”. Therefore, it would be helpful to this reviewer if comparisons between the “before” vs. “1 year” and the “before” vs. “2 year” were made for the data in Figure 2 and Table 3.

Minor issues:

1) The word “especially” can be removed from the last sentence of the abstract.

2) It is stated in the methods that “Care was taken to ensure that vitamin supplementation was carried out regularly”. Could you please provide additional information to describe how this was done?

3) Please confirm the dimensions of the Ascentis C18 column on line 172.

4) In Figure 2, the IgG data should be labeled as panel “D”.

5) It would have been helpful for this reviewer if the normal range of values were described for these parameters.

6) In Table 3, several parameters have “(N = …)” listed next to them. These numbers match those in Table 2 and are greater than 21, but the values shown for the “before start” are different. Is this an error?

7) Are there references available for the statements made in lines 386 – 393?

8) For the statement on lines 387 – 389, it may be better to say that this was beyond the scope of the study and refer readers to Figures 2D and E to say that this trend was observed.

9) Line 440: I believe “intake” should be “plasma levels” or something similar.

10) Line 458: “teczacaftor” should be “tezacaftor”.

Author Response

Response to Reviewer 1

 We thank the reviewer 1 for his/her helpful comments and revised the manuscript accordingly.

 Concerns:

  1. It is stated in the methods that “Variations in the dosage of individual vitamins were only made if the plasma concentrations deviated from the target range…”. Were changes made in the dosage of vitamin supplementation over the course of LUMA/IVA treatment? If so, how common were these changes? If these were common, it would be important to mention this in the manuscript as another potential confounder of the results.

    Answer: We thank the reviewer for this comment and would like to respond as follows:
    A number of internationally consertated guidelines exist for the care of CF patients. There has also been agreement for many years on the amount of lipid-soluble vitamins to be supplemented, as can be seen from the current and previous nutritional guidelines for CF patients (see references below). In this respect, the dose of lipid-soluble vitamins to be supplemented for CF patients has been fixed for many years, and it is evident that so far changes have had to be made in only a few patients. This was also the case in the patients studied here, as a re-examination of the patient data demonstrated. Adjustments were made mostly upward due to age. However, since none of the CF patients so far had values that were too high, no downward adjustment was made for either vitamin A or vitamin E. The situation was different for vitamin D, where seasonal variations and occasionally significantly too low values made adjustments necessary. For this reason, we think that this is unlikely to be an additional potential confounder of our results for retinol and/or vitamin E.
  2.  
  • Turck, D. et al. "Espen-Espghan-Ecfs Guidelines on Nutrition Care for Infants, Children, and Adults with Cystic Fibrosis." Clin Nutr 35, no. 3 (2016): 557-77.
  • Sinaasappel, M. et al. “Nutrition in patients with cystic fibrosis: a European Consensus.” J Cyst Fibros 1 (2002): 51-75.

  1. The statistical analyses performed for Figure 2 and those in the text describing Figure 2 and Table 3 are not clear to this reviewer. In lines 236 – 238, the loss of a significant reduction in LCI over two years is described, but in Figure 2C, I believe that the p value shown is for an ANOVA/Friedman test, which would just describe differences amongst the groups, but these tests would not make comparisons between two groups (unless a post-test was also performed). Were paired t-test/Wilcoxon matched-pairs signed rank tests performed between the “before” vs. “1 year” (and was this significant for the 21 patients?) and also between “before” vs. “2 years” (no longer significant)? I assume this is what was performed for the p values described in lines 271 – 275. While the p values provided by ANOVA/Friedman tests are meaningful to describe general differences amongst groups, it could be that no significant difference was observed between the “before” and “Year 1”, but a difference was observed between “Year 1” and “Year 2”. Therefore, it would be helpful to this reviewer if comparisons between the “before” vs. “1 year” and the “before” vs. “2 year” were made for the data in Figure 2 and Table 3.

    Answer: The reviewer is right in that point. We performed ANOVA/Friedman tests on vitamin concentrations over 2 years and reported the results in Figure 2 and Table 3. However, we also performed Wilcoxon matched-pairs signed rank tests between "before" vs. "1 year" and between "before" vs. "2 years". For LCI, the P-value "before" vs. "1 year" was P=0.0759 and between "before" vs. "2 years" was P=0.1231. Only for the entire cohort of 41 CF patients the difference in LCI "before" vs. "1 year" was significant (see table 2).

Minor issues:

  1. The word “especially” can be removed from the last sentence of the abstract.
    Answer: The word “especially” was removed from the last sentence of the abstract.

  2. It is stated in the methods that “Care was taken to ensure that vitamin supplementation was carried out regularly”. Could you please provide additional information to describe how this was done?
    Answer: The question of the reviewer is very important and we want to answer it as follows: The patients were seen at least four times per year in the outpatient department of the CF center. At each presentation, the intake and tolerability of all medications, including vitamin supplements, are queried. If necessary, the route of intake or dosages are adapted or corrected. In addition, vitamin supplementation preparations had to be prescribed. This enables an approximate estimate of the medications taken to be made during treatment, which is checked on the one hand on a random basis but also in the event of suspected irregularities in intake.

  3. Please confirm the dimensions of the Ascentis C18 column on line 172.
    Answer: The reviewer probably asked this question because he had noticed an error in the specification of the parameters of the HPLC column. In fact, we had subsequently discovered the error. Instead of 3 mm, it should read 2.7 mm, which we corrected accordingly in the manuscript. We therefore thank the reviewer for his attention!
  4. In Figure 2, the IgG data should be labeled as panel “D”.
    Answer:
    We thank the reviewer for this advice! The “IgG” data in Figure 2 were now labeled as panel “D”.

  5. It would have been helpful for this reviewer if the normal range of values were described for these parameters.
    Answer: We thank the reviewer for this comment. As clinicians, we are used to the fact that for most of our clinical parameters there are internationally accepted standard values that do not have to be stated. However, we did not consider that we are not publishing in a primarily clinical journal and that potential readers and reviewers are not familiar with at least some of the parameters we use. We would like to apologize for this! We have therefore decided to describe the normal values in the methods chapter for the lung function values FEV1%pred and LCI, which were probably the trigger for the reviewer's comment. For the other clinical and laboratory parameters, the generally accepted standard values are applicable and, as usual, are not listed separately.

  6. In Table 3, several parameters have “(N = …)” listed next to them. These numbers match those in Table 2 and are greater than 21, but the values shown for the “before start” are different. Is this an error?
    Answer: We thank the reviewer for noticing this error! The subgroup included 21 subjects, of whom values were also available for most of the parameters shown. Only for three of the parameters were fewer than 21 measurements available at one of the time points. According to that, we corrected the values in table 3!
  7.  
  8. Are there references available for the statements made in lines 386 – 393?
    Answer: We thank the reviewer for this question. Yes, there are references for the statements made in lines 386 - 393 about an inverse correlation between plasma vitamin A concentration and proteins describing the state of chronic inflammation. We cited one paper describing the inverse correlation between the plasma concentration of retinol and IgG which was already cited elsewhere in the manuscript and changed the sentence accordingly.

  9. For the statement on lines 387 – 389, it may be better to say that this was beyond the scope of the study and refer readers to Figures 2D and E to say that this trend was observed.
    Answer: We thank the reviewer for this advice and changed the sentence accordingly.

  10. Line 440: I believe “intake” should be “plasma levels” or something similar.
    Answer: Yes, the reviewer is correct! We changed the sentence accordingly.

  11. Line 458: “teczacaftor” should be “tezacaftor”.
    Answer: Yes, the reviewer is correct! We changed the word accordingly.

Reviewer 2 Report

Sommerburg et al present an interesting study examining responses to LUMA/IVA in F508del patients who are PI.  A detailed analysis of clinical outcomes is performed with changes in fat-soluble vitamins A and E and cholesterol being the primary outcomes of focus.  It is determined that serum vitamin A levels rise in response to treatment and that cholesterol levels and vitamin E/cholestserol ratios decline. 

Comments

  1. One postulated mechanism for the decrease in serum cholesterol content is that de novo cholesterol synthesis in the liver, normally elevated in CF, may be decreased in response to treatment.  If samples are still available, serum content of lathosterol has been shown to be an effective surrogate marker of cholesterol synthesis rates and may be useful in directly examining this proposed mechanism. 

  1. Scatter plots in figure 2 show considerable overlap between groups in all categories. It would be useful to see how individual patient responses over time look, particularly with the vitamin A and E data.  For example with the vitamin A measure, is there a consistent rise among most patients or are there some super responders that push the average up while most patients are flat?  It is difficult to interpret the data as presented. 

  1. The significant increase in BMI is assumed to be due to improved digestion and absorption of food. Of course this hypothesis may be true, but a variety of hormonal responses such as improved IGF-1 production and transport and better insulin sensitivity could also significantly contribute to sustained BMI increases over time.

  1. The primary concern of the study is low n’s and no control over diet. Since the lipid changes are rather subtle, it is difficult to ascertain that these changes are due specifically to partial CFTR correction, though the authors do address this point well.

Overall, this is an interesting study that strongly suggests that even less optimal correction of CFTR function with modulators can impact lipid regulation.  There are weaknesses in number of subjects and the influence of diet, but this manuscript could be the basis for further study on the topic. 

Author Response

Response to reviewer 2

 We thank the reviewer 2 for his/her helpful comments and revised the manuscript accordingly.

  1. One postulated mechanism for the decrease in serum cholesterol content is that de novo cholesterol synthesis in the liver, normally elevated in CF, may be decreased in response to treatment.  If samples are still available, serum content of lathosterol has been shown to be an effective surrogate marker of cholesterol synthesis rates and may be useful in directly examining this proposed mechanism. 

    Answer:
    We thank the reviewer for this suggestion. He is correct that the serum level of lathosterol could be an effective surrogate marker for the rate of cholesterol synthesis. However, since all measurements in the study were performed during clinical routine in the laboratories of the university hospital, there were no retained backup samples from these analyses. Therefore, we regret that the proposed measurements cannot be performed anymore.

  2. Scatter plots in figure 2 show considerable overlap between groups in all categories. It would be useful to see how individual patient responses over time look, particularly with the vitamin A and E data.  For example with the vitamin A measure, is there a consistent rise among most patients or are there some super responders that push the average up while most patients are flat?  It is difficult to interpret the data as presented. 

    Answer:
    We thank the reviewer for this comment and agree that it could be possible that there are some super-responders for the individual parameters that drive the average upwards, while the curves of most patients are flat. To better illustrate this, we have therefore produced the figures for retinol and vitamin E/cholesterol suggested by the reviewer (The figures can be found in the pdf-file submitted for reviewer 2). These figures show that for both parameters there is a general trend that can also be seen from the individual curves, even though there are a few patients in whom the trend is not so pronounced. Therefore, we do not share the reviewer's fear that our presentation suggests a general trend that may not even exist, because then the corresponding statistical analyses would not have shown any significant differences either. In order to further support this statement, we have therefore carried out a paired t-test for both retinol and Vit.E/Chol (as each is normally distributed according to the D'Agostino & Pearson omnibus normality test, the Shapiro-Wilk normality test and the KS normality test) for "before" vs. "2 years". These tests also indicated significant differences with P=0.0035 for retinol and P=0.0287 for Vit.E/Chol, which would most likely not have existed if the data had been as the reviewer feared. For this reason, we would want to leave Figure 2 in its current form.
  3. The significant increase in BMI is assumed to be due to improved digestion and absorption of food. Of course this hypothesis may be true, but a variety of hormonal responses such as improved IGF-1 production and transport and better insulin sensitivity could also significantly contribute to sustained BMI increases over time.

    Answer: We thank the reviewer for this valuable comment. We agree that the significant increase in BMI may also be partly due to hormonal responses such as increased IGF-1 production and transport or improved insulin sensitivity. Regarding insulin, there is evidence that insulin production and release is impaired in CF patients due to reduced CFTR activity in the pancreas. Therefore, it is also possible that improved CFTR activity by LUM/IVA leads to improved insulin release. Nevertheless, we believe that improvement in malabsorption of fats is the critical factor leading to an increase in BMI, as conversely malabsorption is known to be the key factor in failure to thrive in CF patients.

  4. The primary concern of the study is low n’s and no control over diet. Since the lipid changes are rather subtle, it is difficult to ascertain that these changes are due specifically to partial CFTR correction, though the authors do address this point well.

    Answer: We thank the reviewer for this comment and agree with him that there are small weaknesses in this real-world study. However, it should not be forgotten at this point that the number of patients with cystic fibrosis who fulfil the requirement for treatment with this CFTR modulator is limited overall. We also thank the reviewer for mentioning our discussion in this context, because a permanent control of the diet of patients over two years in a real-world setting is almost impossible. In this respect, the study was able to provide important insights regarding the course of plasma concentrations of lipid-soluble vitamins, however, as the reviewer correctly noted, these data should only be the starting point for new, more comprehensive, and preferably multicenter projects on this topic.
